# A Liposome-Based Nanoparticle Vaccine Induces Effective Immunity Against EBV Infection

**DOI:** 10.3390/vaccines13040360

**Published:** 2025-03-28

**Authors:** Ping Li, Zihang Yu, Ziyi Jiang, Yike Jiang, Jingjing Shi, Sanyang Han, Lan Ma

**Affiliations:** 1Institute of Biopharmaceutical and Health Engineering, Tsinghua Shenzhen International Graduate School, Tsinghua University, Shenzhen 518055, China; liping@szbl.ac.cn (P.L.); jiang-zy20@mails.tsinghua.edu.cn (Z.J.); shijj21@mails.tsinghua.edu.cn (J.S.); 2Institute of Biomedical Health Technology and Engineering, Shenzhen Bay Laboratory, Shenzhen 518132, China; jiangyk@szbl.ac.cn; 3Institute of Bio-Architecture and Bio-Interactions, Shenzhen Medical Academy of Research and Translation, Shenzhen 518107, China; yuzihang@smart.org.cn; 4State Key Laboratory of Chemical Oncogenomics, Tsinghua Shenzhen International Graduate School, Tsinghua University, Shenzhen 518055, China

**Keywords:** Epstein-Barr virus (EBV), liposome nanoparticle, vaccine, gp350 glycoprotein

## Abstract

**Background:** Epstein-Barr virus (EBV) infects approximately 95% of the global population, causing numerous malignancy-related cases annually and some autoimmune diseases. EBV-encoded gp350, gH, gL, gp42 and gB glycoproteins are identified as antigen candidates for their key role in viral entry, and nanoparticle vaccines displaying them were developed for the advantage of inducing cross-reactive B cell responses. **Methods**: To develop liposomes displaying nanoparticle vaccine, we synthesized liposomes to present the well-identified EBV-encoded gp350D_123_ glycoprotein on their surface to imitate the viral structure, through the conjugation between N-hydroxysuccinimide (NHS) groups on the liposomes and primary amine of antigens to form stable amide bond. Then we assessed the immunogenicity of the biomimetic Lipo-gp350D_123_ nanoparticle vaccine in Balb/c mice immunized experiments. **Results**: The results showed that the sera samples from Lipo-gp350D_123_ nanoparticle vaccine immunized mice collected at weeks 8, 10 and 12 had higher titers of gp350D_123_ protein-specific antibodies, compared to monomer gp350D_123_ protein control, and higher titers of neutralizing antibodies to block EBV-GFP infection in AKATA cells. Meanwhile, the Lipo-gp350D_123_ nanoparticle vaccine also induced higher percentage of CD8+ IFN-γ+ T cells in the spleen, but without significance in CD4+ IFN-γ+ T cells, and these isolated splenocytes showed a higher level of secreted IFN-γ. Moreover, no significant histopathological changes were observed in all vaccinated mice. **Conclusions**: Altogether these data demonstrated that the liposome displaying promoted the immunogenicity of antigens, and the Lipo-gp350D_123_ nanoparticle vaccine candidate had potential application in blocking EBV infection. The liposome nanoparticle was a useful vector for antigen displaying to elicit effective immunity.

## 1. Introduction

Epstein-Barr Virus (EBV) belongs to the gamma herpesvirus family. It is one of the most widespread human viruses with more than 95% of individuals worldwide being infected [1,2,3]. EBV is the first oncogenic virus identified in humans, and has close association with a variety of malignancies, such as nasopharyngeal carcinoma (NPC), gastric carcinoma (GC), Burkitt’s lymphoma (BL), Hodgkin’s lymphoma (HL) and NK/T cell lymphoma, etc. [4,5,6]. EBV was also identified as the major inducement of infection-associated multiple sclerosis [7,8] and ankylosing spondylitis [9,10]. EBV can also infect NK/T cells to form a natural killer/T cell lymphoma [11]. A study in 2008 reported that there were approximately 200,000 EBV-associated cancer cases and 140,000 deaths annually around the world [12,13], causing heavy global burden.

EBV virions are primarily transmitted through saliva and infect the epithelial cells of the oropharynx and resting B lymphocytes, where they establish a life-long latent infection. These infected B lymphocytes also carry the virus to lymphoid tissues and peripheral blood, thereby expanding the infection [14]. During latent infection, EBV-encoded proteins, such as EBNA1, EBNA3A, EBNA3C and LMP1 et al., dysregulate host immunity responses [7,8,9,10]. Most of these EBV-infected cells are in the latent stage, but they could switch into the replication phase to produce and release progeny viruses under some stimulations. This reactivation is driven by the viral BZLF1 protein, which can be stimulated by some chemical or biological inducers, such as 12-*O*-tetradecanoylphorbol-13-acetate, calcium ionophore or histone deacetylase inhibitors [15]. The lytic cycle of EBV plays a pivotal role in development and maintenance of tumors by promoting the expression of some cytokines or growth factors to induce inflammation and angiogenesis [16]. Blocking EBV infection at the primary stage by prophylactic vaccine to prevent EBV-associated disease is needed, but no vaccine is approved for clinical use at present.

The viral particles of EBV feature multiple envelope glycoproteins on their surface, mainly including gp350, gL, gH, gp42 and gB, which mediate the adhesion and invasion of virus to host cells through cooperation [17]. Among them, the gp350 protein is the most abundant glycoprotein on the surface of EBV virions and interacts with the complement receptor 2 (CR2/CD21) on B lymphocytes to trigger viral infection [18]. The monoclonal antibody 72A1 isolated and identified from gp350 protein immunization mice could efficiently inhibit EBV infection in vitro and in vivo [19,20]. Thus, it was well-studied and identified in prophylactic subunit vaccines to induce neutralizing antibodies [19,21,22,23,24]. In addition, the gL, gH and gp42 proteins were identified for mediating the EBV infection in epithelial cells, and vaccines based on them also could elicit robust neutralizing antibody response against EBV infection [25]. The EBV-encoded gB glycoprotein is responsible for the core membrane fusion through conformational change [26,27]. Recently, two gB-specific neutralizing antibodies, 3A3 and 3A5, were isolated and shown to effectively neutralize the dual-tropic EBV infection of B lymphocytes and epithelial cells [28], and a gB protein-based nanoparticle vaccine provided efficient protection against lethal EBV challenge [29]. In addition, two mRNA vaccines against EBV constructed by Moderna, mRNA-1189 (gL, gH, gp42 and gp220) and mRNA-1195 (gLgH, gp42, gp220 and latent protein), are being evaluated in clinical phase 1/2 trials. The mRNA-1189 vaccine clinical trial (NCT05164094) was reported as complete recently, and the report showed a significant inhibition of viral load in saliva in mRNA-1195-immunized subjects, compared to placebo-controlled. These studies demonstrated that the gp350, gL, gH, gp42 and gB glycoproteins are crucial for vaccine design to provide adequate protection against EBV infection.

Nanoparticle vaccines were developed in EBV prophylactic vaccines for the advantage of inducing cross-reactive B cell responses, showing an avidity advantage over strain-specific B-cell receptor (BCR) interactions [30]. These virus-like particle (VLP) vaccines based on ferritin, I53-50A1 protein and hepatitis B core antigen (HBc149) showed advantages in more efficient antigen presentation, higher specific antibody titers and stronger protection [29,31,32,33]. Among them, the EBV gp350-ferritin vaccine constructed by the National Institute of Allergy and Infectious Diseases is undergoing clinical phase 1 studies, including NCT04645147 and NCT05683834. Recently, Zhong et al. formulated three nanovaccines using a core-shell structure liposome with DOTAP, PLGA and DSPE-PEG2000 to package molecular adjuvants (CpG and MPLA) and antigens (gHgL, gB or gp42), which induced robust humoral and cellular responses [34]. Coincidentally, we also investigated a nanovaccine based on a liposome, but it was used for displaying antigens to simulate the viral structure.

In our study, we synthetized some liposomes as fusion apparatus to display antigens on their surface through the N-hydroxysuccinimide (NHS) groups exposed on liposome surfaces to conjugate with a primary amine on antigen proteins to form stable amide bonds. Then, these liposomes were applied to the present gp350D_123_ protein, a truncated gp350 protein (1–425 aa) including the receptor binding domain (RBD). To investigate the immunogenicity of the biomimetic liposome-based nanoparticle vaccine, a mouse immunization assay was performed and the results showed that the liposome nanoparticle vaccine elicited a higher antigen-specific antibody titer and neutralizing antibody titer, and also induced a higher percentage of CD8+ IFN-γ+ T cells and significant increase of secreted IFN-γ in splenocytes, compared with the monomer gp350D_123_ protein control. These results demonstrated that this liposome-based nanoparticle provided a valuable platform to present EBV antigens and elicit a stronger response against EBV infection.

## 2. Materials and Methods

### 2.1. Cell Culture

HEK293 cells were cultured in suspension medium (KOP293, KAIRUI BIOTECH, Zhuhai, China) at 37 °C with 5% CO_2_ and shaking. CNE2 cells (EBV transformed human nasopharyngeal carcinoma cell line, CVCL_6889) and AKATA cells (EBV transformed Burkitt’s lymphoma cell line, CVCL_3080) were cultured in RPMI-1640 medium and supplemented with 10% fetal bovine serum (FBS) at 37 C with 5% CO_2_. Cell lines were tested for mycoplasma.

### 2.2. Plasmids Construction and Recombinant Protein Expression and Purification

The EBV gp350D_123_ glycoprotein gene was constructed with an IL-2 secreted peptide at the N-terminus. Genes were cloned into pLVX-IRSE-GFP-Puro and confirmed by sequencing. The plasmid was transfected into HEK293 cells for expression and secretion of the soluble gp350D_123_ protein. The cell supernatant with gp350D_123_ glycoprotein was collected and then purified in an AKATA Protein Purification System with Ni-NTA (GE healthcare, Chicago, IL, USA) and in a Superose 6 Increase 10/300 GL gel filtration column (GE healthcare). The protein was determined with a BCA kit (Solarbio, Beijing, China) and identified by WB.

### 2.3. Native-PAGE, SDS-PAGE and Western Blotting (WB)

The migration of protein and liposome-protein samples was performed in 4% polyacrylamide gel electrophoresis without sodium dodecyl sulfate (SDS) (native-PAGE) and then migrated in TBE buffer. The Coomassie blue buffer was used to stain the whole gel to analyze the binding of liposomes and gp350D_123_ glycoprotein. The boiled samples were identified by SDS-PAGE assay. Then, proteins were transferred to PVDF membranes (Millipore, Burlington, MA, USA) and treated with blocking buffer containing 5% bovine serum albumin (BSA), and incubated with gp350-specific or tag-specific antibodies, and corresponding HRP-conjugated-secondary antibody. The results were analyzed in a chemiluminescence imaging system (BioRAD, Hercules, CA, USA).

### 2.4. Synthesis of Liposomes, and Assembly with Antigen Proteins

Liposomes were prepared and conjugated to purified protein gp350D_123_. Briefly, liposomes composed of cholesterol, 1,2-dioleoyl-sn-glycero-3-phosphoethanolamine (DOPE), 1,2-distearoyl-sn-glycero-3-phosphorylcholine (DSPC) and DSPE-PEG2000-NHS were prepared at different molar ratios. The lipid mixture (10 mg) was dissolved in a round-bottomed flask using a chloroform solvent, and then was removed into a rotary evaporator (N-1200AV; EYELA, Tokyo, Japan), generating a thin-layer lipid film. The dried lipid films were hydrated in 10 mL of PBS solvent in atmospheric pressure for 2–3 h. Liposomes were extruded through a polycarbonate membrane with pore sizes of 100 nm (610005-1Ea, Avanti, Weston, FL, USA) using an Avanti extrusion device (610017-1Ea, Avanti, USA).

Synthetic liposomes were mixed with purified recombinant protein gp350D_123_ (1:10, molar ratio of gp350D_123_ and liposome), and then incubated overnight at 4 °C with continuous rotating. The mixture was further concentrated and purified by ultrafiltration and Superose 6 Increase 10/300 GL gel filtration column (GE healthcare). The liposome-gp350D_123_ conjugations were further determined by BCA kit (Solarbio) and WB, and then they were stored at 4 °C.

### 2.5. Attenuated Total Reflectance (ATR) and Dynamic Light Scattering (DLS) Detection

Liposomes and gp350D_123_ conjugations (Lipo-gp350D_123_) were characterized using attenuated total reflectance (ATR) and dynamic light scattering (DLS) assays. For ATR detection, liposome and Lipo-gp350D_123_ nanoparticles were frozen at −80 °C overnight, and then the samples were frozen for drying at −40 °C in a vacuum environment using a freeze drier (SCIENTZ-10N/C, SCIENTZ, Ningbo, China) to obtain powdered samples. The samples were characterized through ATR detection using a Fourier-transform infrared spectrometer (Frontier, PerkinElmer, Waltham, MA, USA).

The DLS was used for the detection of nanoparticle size. The intensity of freshly synthesized liposome and Lipo-gp350D_123_ nanoparticle samples were measured using a nanoparticle size analyzer (Nano ZS90, Malvern, UK).

### 2.6. Transmission Electron Microscopy (TEM)

Negative staining electron microscopy was performed to analyze the structure of liposomes and Lipo-gp350D_123_ nanoparticles. Briefly, we diluted these samples to 0.5 mg/mL and then added the samples to carbon-coated copper grids (200-mesh) for 5 min. We washed the grids 3 times with ddH_2_O and then performed the negative staining with 2% phosphotungstic acid with a pH of 6.4 for 30 s. The imaging was collected by a FEI Tecnai T12 TEM (FEI, Hillsboro, OR, USA) at an accelerating voltage of 120 kV and photographed at a magnification of 25,000-fold.

### 2.7. Immunization Assay

The mice-associated experiments in this study were approved by the Institutional Animal Care and Use Committee (IACUC) of Tsinghua Shenzhen International Graduate School. Special pathogen-free (SPF) mice were purchased from ZhuHai Bestest Biotechnology Co., Ltd. (Zhuhai, China), and were fed in accordance with the institutional guidelines.

Six-week-old SPF female Balb/c mice, eight per group, were immunized by intramuscular injection (i.m.). Purified gp350D_123_ protein (about 15 μg) and Lipo-gp350D_123_ (protein equivalent, about 15 μg) were diluted in PBS and then injected at week 0 (prime), week 2 (boost 1) and week 4 (boost 2). The sera of all immunized mice were separately collected from week 0 to week 14, and stored at −80 °C prior to use to detect the protein-specific IgG titer and neutralization antibody titer. Euthanasia was finally performed at week 14 to collect the major tissues of immunized mice, including lung, spleen, heart, liver and kidney, for histopathology analysis. The rest of the spleen of each mouse was used for splenocyte isolation.

### 2.8. Splenocyte Isolation and Intracellular Cytokine Staining (ICCS)

We isolated the splenocytes from mice as described previously [31]. Briefly, we collected the spleens from immunized mice and washed them with sterilized PBS. The spleens were homogenized with a cell strainer (70 μm, Corning, Corning, NY, USA). The single cells were then suspended with ACK lysis buffer (Elabscience, Houston, TX, USA) for 5 min to remove red blood cells. Splenocytes were directly stained with fluorescence-conjugated monoclonal antibodies against surface markers, including anti-CD3-PE, anti-CD4-BV510 and anti-CD8-FITC (Elabscience), in PBS buffer (containing 0.5% BSA). To stain intracellular proteins, cells were fixed in 4% paraformaldehyde for 10 min, and then permeabilized with 0.25% Triton X-100 for 10 min. The intracellular cytokines were stained with fluorescence-conjugated monoclonal antibodies (anti-IFN-γ-APC, Elabscience) for 30 min on ice. These stained cells were then detected by flow cytometry.

### 2.9. Indirect Enzyme-Linked Immunosorbent Assay (ELISA)

We coated gp350D_123_ protein onto 96-well microplates (Jet Bio, Guangzhou, China), 200 ng per well, in the CBS buffer (0.015 M Na_2_CO_3_, 0.035 M NaHCO_3_, pH 9.6) and left them overnight at 4 °C. Then, we washed the microplates 3 times and blocked the sample wells with blocking buffer (0.02 M PBS pH 7.4, 0.05%Tween20, 1% BSA) for 1 h at 37 °C. After washing the microplates 3 times, we added the mouse sera samples with 10-fold serial dilution into the well and incubated them for 1 h at 37 °C. Next, we washed the microplates 3 times and added secondary antibody horseradish peroxidase (HRP)-conjugated goat anti-mouse IgG (Solarbio) (1:4000 dilution) into the wells to incubate for 1 h at 37 °C. The final signals were developed in buffer (0.0243 M citric acid, 0.0514 M Disodium hydrogen phosphate, 0.045% H_2_O_2_, 0.0037 M OPD) for 10–20 min. We stopped the reaction using 2 M H_2_SO_4_ and detected the absorbance at 492 nm within 5 min with a microplate reader (Molecular Devices, San Jose, CA, USA).

### 2.10. Production and Purification of EBV-GFP Reporter Virus

The EBV-GFP reporter virus (EBV-GFP) induction, purification and infection assay was performed according to the method previously reported [35]. The EBV-GFP reporter virus was produced from CNE2-EBV-GFP cells. Briefly, CNE2-EBV-GFP cells were cultured in T75 flasks and about 12 h post-cell passage, cells were treated with 12-o-tetradecanoylphorbol 13-acetate (20 ng/mL) and sodium butyrate (2.5 mM) for 24 h, so that the EBV genome would be activated to produce and release virus into the medium. We collected the supernatant of the cell culture and filtered it with 0.8 μm filters (Sartorius, Bohemia, NY, USA). After ultracentrifugation at 50,000× *g* for 2.5 h at 4 °C, the white pellet on the bottom was collected and gently resuspend in fresh medium, and then stored at −80 °C.

EBV-GFP infection of EBV-negative AKATA cells was carried out to detect the viral infection efficiency. The EBV-GFP was diluted and incubated with 10^4^ AKATA cells per well in 96-well plates for 2 h at 37 °C. After washing with PBS, these cells were then cultured for 36 h at 37 °C. The infection rate of EBV-GFP reporter virus was measured by flow cytometry to detect the percentage of GFP+ cells.

### 2.11. EBV Infection Blocking Experiment

The EBV-GFP reporter virus (EBV-GFP) was used in this study for the infection blocking experiment according to the method used in a previous study [29]. We collected sera samples from all Balb/c mice, diluted them with 10-fold serial dilution and mixed this with 50 μL EBV-GFP solution to incubate for 2 h at 37 °C. Subsequently, the sera–virus mixtures were added to 10^5^ EBV-negative AKATA cells and incubated at 37 °C for 3 h. After incubation, cells were washed with PBS, then cultured in fresh medium with 10% FBS in 24-well plates for 48 h. Before detection, the cells were washed 3 times and resuspend in PBS. The GFP-positive cells were determined by flow cytometry. The results were analyzed by FlowJo 10.8.1 software.

### 2.12. Histopathology Analysis

All the immunized Balb/c mice were euthanized at the endpoint of experiments. We collected their major tissues, including lung, spleen, heart, liver and kidney, and then fixed them in 4% paraformaldehyde for 48 h. The fixed tissues were embedded in paraffin and cut into longitudinal sections and stained with hematoxylin and eosin (H&E). Finally, the images were captured with a Pannoramic MIDI device (3DHISTECH Ltd., Budapest, Hungary).

### 2.13. Data Statistics

Data collected and used in this study were subject to statistical analysis, and the details are described in figure legends. The statistical significance was calculated by two-tailed unpaired Student *t*-test and a *p* value < 0.05 was deemed significant. All the data are shown as mean ± SD and were analyzed by GraphPad Prism 8.3 software.

## 3. Results

### 3.1. Rational Molecular Design of Liposome-Based Vaccine

In this study, the liposomes were chosen for antigen protein presentation on the surface to simulate the viral particles. These liposomes were composed of cholesterol, DOPE, DSPC and DSPE-PEG2000-NHS with different molar ratios. Through multi-step synthesis, liposomes with N-hydroxysuccinimide (NHS) groups exposed on their surface were produced. These NHS groups functioned as anchors to conjugate with primary amines on antigen proteins to form stable amide bonds, and then displayed them on the surface of liposomes after co-incubating at 4 °C overnight (Figure 1A). The well-studied EBV-encoded gp350 glycoprotein was designed as the antigen in this study. A truncated gp350 protein (1–425 aa) covering the receptor binding domain (RBD), named gp350D_123_, was cloned, expressed and purified (Figure 1B). The results in Figure 1C show that the purified gp350D_123_ protein could be identified by gp350-specific antibody. To improve the combination rate of the antigen protein on liposomes, the formulation of liposomes was optimized, and the four reagents were mixed at different molar ratios to produce five kinds of liposomes (Figure 1D). The purified gp350D_123_ proteins were then incubated with these liposomes and a Native-PAGE electrophoresis assay, along with Coomassie blue staining, was performed to detect the binding of liposomes and proteins. As shown in Figure 1E, the mobility of the gp350D_123_ protein in the gel was apparently retarded by the liposomes, indicating the formation of the liposome-gp350D_123_ protein complex. Liposome 5 showed an optimum binding to the gp350D_123_ protein, with less protein migrating into the gel under the conditions of an equal amount of protein compared to other liposomes. Thus, we used Liposome 5 for further vaccine construction.

### 3.2. Characterization of Lipo-gp350D_123_ Nanoparticle

The Lipo-gp350D_123_ nanoparticles based on Liposome 5 were constructed as our vaccine candidate and the characterization was investigated. Firstly, the retention of Lipo-gp350D_123_ and liposomes by size exclusion chromatography (SEC) was detected. Usually, pure liposomes have no absorbance at 280 nm, but the NHS groups in DSPE-PEG2000-NHS showed some absorbance values (Figure 2A). As Figure 2A shows, Lipo-gp350D_123_ had a significantly higher absorbance at 280 nm, indicating the conjunction of protein with liposome. Furthermore, the retention volume showed that Lipo-gp350D_123_ had a smaller retention peak at 7.85 mL, compare to liposomes at 8.11 mL. These data indicated the successful conjugation of gp350D_123_ to liposomes. Subsequently, we detected the gp350D_123_ protein on the nanoparticles by SDS-PAGE electrophoresis with Coomassie blue staining and Western-blot analysis, and the results displayed an apparent enrichment of gp350D_123_ protein in Lipo-gp350D_123_ nanoparticles (Figure 2B). Then, the liposomes and Lipo-gp350D_123_ nanoparticles were characterized via an infrared spectra analysis, and the spectral data of Lipo-gp350D_123_ nanoparticles, presented in Figure 2C, revealed two additional and evident absorption peaks compared with unconjugated liposomes at 1550 cm^−1^ and 1634 cm^−1^, which were characteristic peaks of the amide bonds, suggesting that the gp350D_123_ protein was successfully conjugated to liposomes through the reaction of N-hydroxysuccinimide and primary amine. In addition, the liposomes and Lipo-gp350D_123_ nanoparticles were characterized by transmission electron microscopy (TEM) with negative staining. The phosphotungstic acid bound to the hydrophilic lipid heads in liposomes and created two dark lines with a light intermediate layer, which was the bilayer of liposomes. Figure 2D displays the TEM images showing that these liposome nanoparticles had a thick bilayer structure, but a thin layer between the layers; thus, the liposomes showed a three layer structure. The Lipo-gp350D_123_ nanoparticles were covered and surrounded by a darker stained layer, indicating the binding and displaying of antigen proteins. Finally, we detected the size of these nanoparticles by dynamic light scattering (DLS). The results showed that the nanoparticles in both of the two groups had a single peak within size distribution, and compared with 184.183 nm of liposome nanoparticles, Lipo-gp350D_123_ nanoparticles had a larger particle size of 201.137 nm (Figure 2E). These data indicated that the gp350D_123_ protein was successfully assembled on the surface of liposome nanoparticles to construct the Lipo-gp350D_123_ nanoparticle vaccine.

### 3.3. Immunogenicity of the Lipo-gp350D_123_ Nanoparticle Vaccine

To assess the immunogenicity of the Lipo-gp350D_123_ nanoparticle, 6-week-old Balb/c mice (*n* = 8 per group) were vaccinated with PBS, liposome, gp350D_123_ protein and Lipo-gp350D_123_ nanoparticles, respectively, three times (prime, boost 1 and boost 2) at 2-week intervals. All mouse sera were collected biweekly and they were euthanized at week 14. The mouse splenocytes were isolated and stained for analysis of cytokines and their main tissues were collected for histopathology analysis (Figure 3A). The gp350D_123_ protein-specific antibodies were determined by ELISA and the results showed that both of Lipo-gp350D_123_ nanoparticle and gp350D_123_ protein-vaccinated mice had their highest titer at week 10, and the Lipo-gp350D_123_ nanoparticles elicited significantly higher antibody titers than the monomer gp350D_123_ protein control during weeks 4 to week 14 (Figure 3B). Furthermore, the splenocytes were isolated and stimulated with gp350D_123_ peptide pool and stained with different fluorescence-conjugated antibodies to perform the intracellular cytokine staining (ICCS) (Figure 3C). As shown in Figure 3D,E, the Lipo-gp350D_123_ nanoparticle-immunized Balb/c mice had a higher percentage of CD8+ IFN-γ+ T cells (Figure 3D), indicating a stronger CD8+ dependent T cell response. Comparison between the percentages of CD4+ IFN-γ+ T cells of Lipo-gp350D_123_ nanoparticles and monomer gp350D_123_ protein showed no significance (Figure 3E), indicating that the Lipo-gp350D_123_ nanoparticles had little effect on CD4+ dependent T cell response. Meanwhile, the IFN-γ secreted by these stimulated splenocytes was detected by ELISA and, as shown in Figure 3F, the Lipo-gp350D_123_ nanoparticles induced a higher level of IFN-γ than the monomer gp350D_123_ protein control. These results indicated that the Lipo-gp350D_123_ nanoparticle vaccine elicited an enhanced response in gp350D_123_-specific antibody and CD8+ dependent T cell response, compared with the monomer gp350D_123_ protein.

### 3.4. Lipo-gp350D_123_ Nanoparticle Vaccine Induced Effective Neutralizing Antibody Response

To further evaluate the functional neutralizing capability of the Lipo-gp350D_123_ nanoparticle vaccine, we conducted an in vitro neutralization assay in AKATA cells by detecting the blocking efficiency of sera samples from immunized mice. We used EBV-GFP reporter virus to perform the blocking experiments, which was inserted in a GFP gene in the viral genome.

The EBV-GFP-infected cells (GFP-positive cells) were determined by flow cytometry analysis and approximately 16% of AKATA cells were infected in the absence of antibodies (Figure 4A). Subsequently, we diluted sera from week 10 with 2-fold dilutions and the results showed that the sera samples from Lipo-gp350D_123_ nanoparticle-immunized mice had stronger neutralizing efficiency, compared to gp350D_123_ protein control, and the corresponding ID_50_ (half-maximal inhibitory dilution) values were 19.68 ± 0.89 and 10.95 ± 0.75, respectively (Figure 4B). We also analyzed the sera samples from week 8 and week 12 with 2-fold dilutions. The results showed that the Lipo-gp350D_123_ nanoparticles induced stronger neutralizing efficiency from week 8 to week 12, with the ID_50_ values of 11.78 ± 1.06 (week 8) and 10.35 ± 0.81 (week 12) in Lipo-gp350D_123_ nanoparticle-immunized mice, and 7.19 ± 1.07 (week 8) and 5.4 ± 0.65 (week 12) in gp350D_123_ protein-immunized mice (Figure 4C). These data demonstrated that the Lipo-gp350D_123_ nanoparticle vaccine induced an effective neutralizing antibody response against EBV infection.

### 3.5. Lipo-gp350D_123_ Nanoparticle Vaccine Demonstrates Favorable Safety in Balb/c Mice

At the conclusion of the study, to investigate the safety of the Lipo-gp350D_123_ nanoparticle vaccine, we collected the main tissues of immunized mice, including the lungs, heart, liver, spleen and kidneys, to perform histopathological analysis. As shown in Figure 5, tissues of immunized mice from the four groups showed no significant pathological changes through histological sectioning and H&E staining, which indicated that the Lipo-gp350D_123_ nanoparticle vaccine exhibited favorable safety.

## 4. Discussion

EBV is the first tumorigenic virus identified in humans and causes various malignancies and EBV-associated disease. Currently, numerous preventive and therapeutic approaches are under development in either clinical trials or preclinical research phases. However, no drug, vaccine or therapy specifically targeting EBV infection has yet been approved. In this study, we developed a liposome-displaying platform to present EBV-encoded gp350D_123_ glycoprotein on its surface to construct a lipo-gp350D_123_ nanoparticle vaccine. The mouse vaccinated assay investigating the immunogenicity of the lipo-gp350D_123_ nanoparticle vaccine showed that the vaccine could effectively activate the B-cell immune response with a high titer of gp350D_123_ protein-specific antibody and neutralizing antibody to block EBV-GFP infection in an in vitro cell model, and also activate the CD8+ IFN-γ+ based T-cell immune response. Our findings suggest that the liposome nanoparticle vaccine displaying the gp350D_123_ protein could be an attractive strategy for blocking EBV infection, and further controlling the EBV-associated disease.

In recent years, liposome-based lipid nanoparticles (LNPs) have been widely used in mRNA vaccines against viruses and tumors [36,37,38]. The successful application of two mRNA vaccines against SARS-CoV-2, BNT162b2 (Pfizer/BioNTech) and mRNA-1273 (Moderna) demonstrated the surprising safety and delivery effectiveness of LNPs [39,40]. In these vaccines, the LNPs were usually used to package and encapsulate the mRNA inside the liposomes to protect these cargos from degradation by the host and deliver them into the cells for intracellular release. Actually, peptides were often modified on the surface of liposomes for cell targeting and stimulation [41,42,43]. Antigen proteins displaying on the surface of liposomes could simulate the structure of virus and be recognized by host lymphocytes. In this study, we synthetized the liposomes with cholesterol, DOPE, DSPC and DSPE-PEG2000-NHS, and the N-hydroxysuccinimide (NHS) groups were exposed on the surface of liposomes and functioned as anchors to conjugate with primary amines on antigen proteins to form stable amide bonds, and then displayed them on the surface of liposomes (Figure 1A). The stable amide bond in the lipo-gp350D_123_ nanoparticle vaccine could be detected by attenuated total reflectance assay, demonstrating the successful binding of antigens on the surface of liposomes. The TEM images also showed that the lipo-gp350D_123_ nanoparticles maintained a complete lipid bilayer structure after incubated with antigens overnight with rotating and ultrafiltration, indicating the structural stability of the liposomes. In addition, these vaccinated mice showed no significant of pathological changes in their main tissues, indicating the safety of the liposome vaccine. Thus, our study provided an instance showing that liposomes have potential application in nanoparticle vaccines.

EBV-infected epithelial cells and B lymphocytes rely on the co-operation of multiple EBV-encoded envelope glycoproteins, including gp350, gL, gH, gp42 and gB, and they have been well-identified in vaccine studies. Currently, several EBV vaccines are in phase I clinical trials to evaluate the safety and immunogenicity (initiated by Moderna and the National Institute of Allergy and Infectious Diseases (NIAID)). The NIAID developed a gp350-ferritin nanoparticle vaccine, and studies NCT04645147 and NCT05683834 have been initiated to assess the gp350 glycoprotein-ferritin nanoparticle combined with an adjuvant in healthy adults with or without EBV infection [44]. Meanwhile, the mRNA vaccine mRNA-1189 (Moderna, NCT05164094) was designed based on four glycoprotein antigens of EBV (gH, gL, gp42 and gp350), and a clinical trial was conducted in 10- to 30-year-old healthy adolescents and adults [45]. To provide complete protection against EBV infection, multiple envelope glycoproteins should be combined in vaccines to induce a stronger neutralizing antibody response. Thus, research on fusion apparatus to display multi antigens is necessary.

Liposomes are usually used to package cargos for intracellular delivery. The packaging strategy protects antigens from degradation and efficiently activates dendritic cells and germinal center response in collaboration with adjuvants [34]. In this study, our displaying strategy increased the molecular weight of antigens to enhance their stability, and the covalent antigen conjugation ensured precise spatial orientation for B-cell recognition. In addition, compared to other nanovaccines with protein-derived fusion apparatus, such as ferritin and hepatitis B core antigen, liposomal formulations are cost-effective and easier to manufacture. This study demonstrated the enhancing effect of the liposome nanoparticle platform in antigen protein immunogenicity. Liposome nanoparticles displaying gp350, gH, gL, gp42 and gB glycoproteins will be developed and challenged in EBV-infected humanized mouse and non-human primate models for a comprehensive evaluation.

## 5. Conclusions

In conclusion, this study provides some encouraging data of a novel EBV vaccine candidate based on liposome nanoparticles displaying gp350 glycoprotein antigens. The liposome displaying strategies help improve the immunogenicity of antigens, indicating their potential application in vaccine design. Our further research will try to construct multiple-target vaccines, including gp350, gH, gL, gp42 and gB, based on the liposome platform and also perform an evaluation in humanized mouse and non-human primate models. Overall, our work provides a new strategy of EBV vaccine design.

## Figures and Tables

**Figure 1 vaccines-13-00360-f001:**
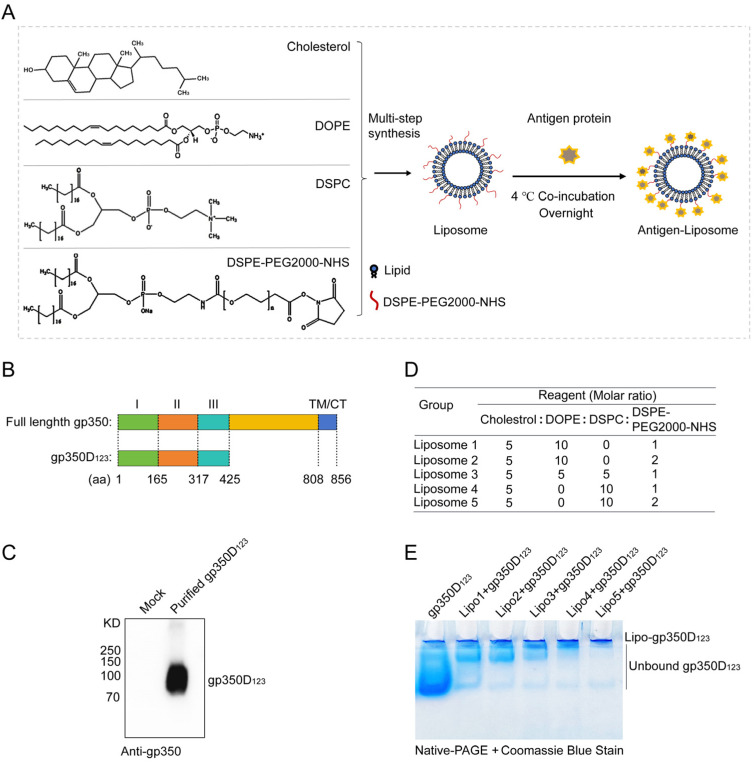
Schematic diagram of the rational design and synthesis of liposome-based vaccine. (**A**) Schematic diagram showing the components of liposomes and binding with antigen proteins. (**B**) Schematic diagram showing the constructing strategy for gp350D_123_ protein. The full-length gp350 protein including three independent domains I (green), II (orange), and III (azure); a undefined region (yellow); and a ectodomain and transmembrane (TM)/cytoplasmic tail (CT) (blue). (**C**) Western-blot detecting the purification of gp350D_123_ protein with gp350-specific antibody. (**D**) The construction formula of liposomes. Cholesterol, DOPE, DSPC and DSPE-PEG2000-NHS were mixed with different molar ratios. (**E**) Native-PAGE detection the binding of gp350D_123_ protein with liposome candidates. Liposome and gp350D_123_ protein were incubated in PBS buffer and fractionated by 4% Native-PAGE, then the gel was stained with Coomassie blue buffer.

**Figure 2 vaccines-13-00360-f002:**
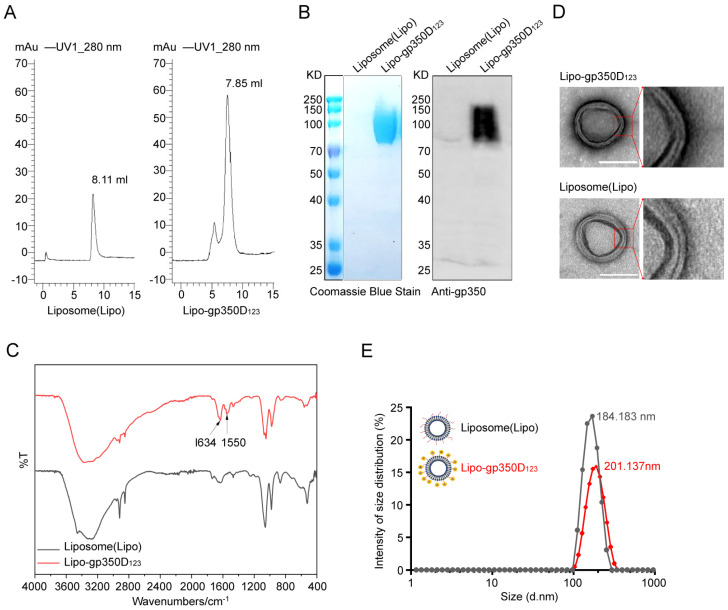
Characterizing assembly of the gp350D_123_ protein on the liposome. (**A**) Retention of liposomes and Lipo-gp350D_123_ by size exclusion chromatography (SEC). (**B**) SDS-PAGE with Coomassie blue staining and Western-blot analysis with gp350-specific antibody detecting the gp350D_123_ protein in Lipo-gp350D_123_. (**C**) Attenuated total reflectance (ATR) assay detecting the stable amide bond in the liposomes (gray) and Lipo-gp350D_123_ (red). The arrows show the additional and evident absorption peaks at 1550 cm^−1^ and 1634 cm^−1^. (**D**) The TEM with negative staining characterizing the structure of liposomes and Lipo-gp350D_123_ nanoparticles. Scale bars represent 100 nm. (**E**) Size analysis of liposomes (grey) and Lipo-gp350D_123_ (red) by dynamic light scattering (DLS).

**Figure 3 vaccines-13-00360-f003:**
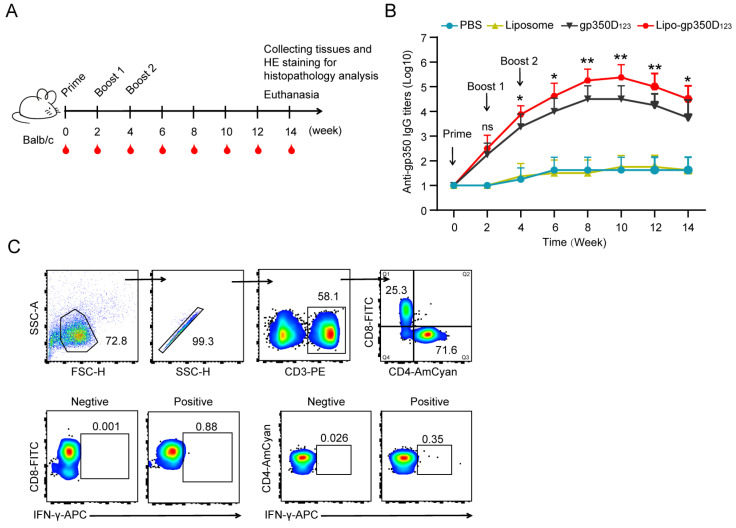
Immunogenicity of Lipo-gp350D_123_ vaccine in Balb/c mice. (**A**) Schematic diagram showing the vaccination experiments in Balb/c mice. SPF mice in each group were immunized at week 0, 2 and 4 and euthanized at week 14. Mouse sera were collected biweekly and their splenocytes were isolated for analysis of intracellular cytokines staining. (**B**) The anti-gp350D_123_ protein IgG titers of all the mice were detected by ELISA. Mouse IgG titers were determined by 10-fold serial dilution. (**C**) Gating strategy for surface marker staining (anti-CD3-PE, anti-CD4-BV510 and anti-CD8-FITC) and intracellular cytokine staining (anti-IFN-γ-APC) in CD8+ and CD4+ T cells, CD8+ or CD4+ T cells as indicated. Intracellular cytokine gating example is collected from a mouse in PBS group as negative control and selected a representative T cell cytokine response to antigens as positive control. (**D**,**F**) Splenocytes were incubated with gp350D_123_ protein peptides pool. The percentages of CD8+ IFN-γ+ (**D**) or CD4+ IFN-γ+ (**E**) T cells were determined by ICCS. The IFN-γ secreted by the stimulated splenocytes was detected by ELISA (**F**). *n* = 8 biological replicates. Statistical analysis with two-tailed unpaired Student’s *t*-test. Data are mean ± SD. * *p* < 0.05, ** *p* < 0.01, and **** *p* < 0.0001, and ns = not significant.

**Figure 4 vaccines-13-00360-f004:**
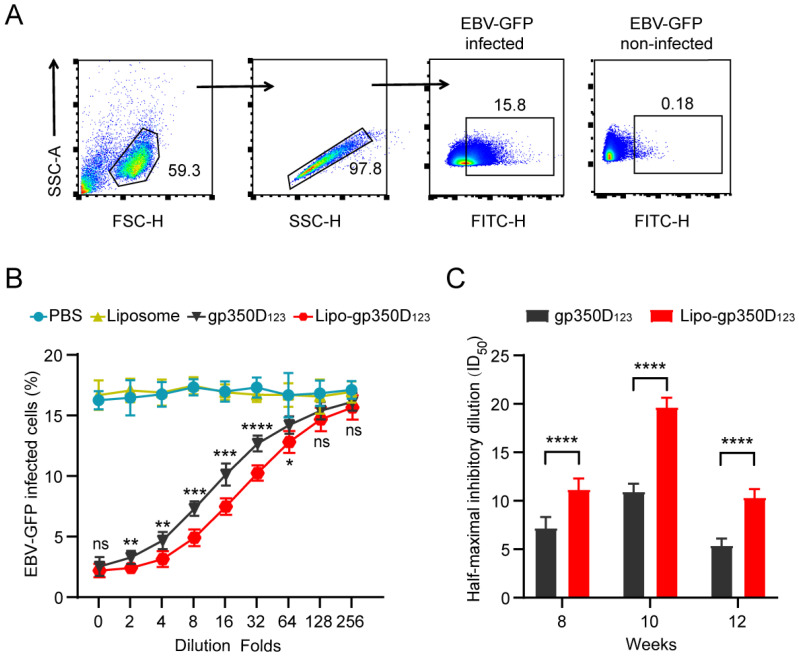
Neutralization of EBV infection. (**A**) Gating strategy for detecting EBV-GFP infected AKATA cells. AKATA cells infected by EBV-GFP or PBS were determined by flow cytometry. (**B**,**C**) Blocking EBV infection by sera samples of immunized mice. Sera samples from immunized mice at week 10 were serially diluted to block EBV-GFP infection in AKATA cells (**B**). The half-maximal inhibitory dilution (ID_50_) was used to investigate the neutralization of sera samples from immunized mice at week 8, 10 and 12 (**C**). Significance between group of gp350D_123_ protein and Lipo-gp350D_123_ nanoparticle vaccine was calculated. *n*= 8 biological replicates. Statistical analysis with two-tailed unpaired Student’s *t*-test. Data are mean ± SD. * *p* < 0.05, ** *p* < 0.01, *** *p* < 0.001, **** *p* < 0.0001, and ns = not significant.

**Figure 5 vaccines-13-00360-f005:**
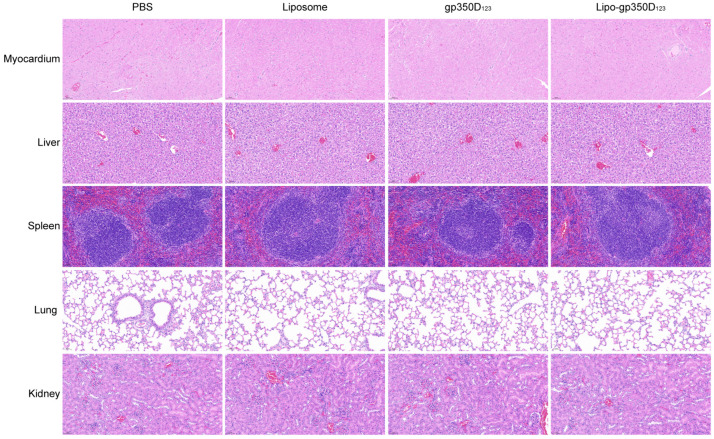
Histopathological analysis of tissues from vaccinated mice. All immunized mice were euthanized and their main tissues were collected immediately to be fixed, sliced and stained by H&E for histopathological analysis, including myocardium, liver, spleen, lung and kidney. Scale bars represented as 50 μm.

## Data Availability

The original contributions presented in this study are included in the article/Appendix A. Further inquiries can be directed to the corresponding authors.

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
