# Peer review of "A Liposome-Based Nanoparticle Vaccine Induces Effective Immunity Against EBV Infection"

_vaccines, 2025, doi:10.3390/vaccines13040360_

Round 1
Reviewer 1 Report
Comments and Suggestions for Authors
The EBV virus, which inhabits the lymphocytes of more than 90% of the Earth's population, has been actively studied for more than 50 years, however, new features of its interaction with the human body and the pathogenesis of a wide range of diseases associated with this virus are constantly being identified. Lymphocytes are the primary target of the Epstein-Barr virus, so infection with it causes serious damage to the immune system and makes the creation of an effective preventive vaccine remains illusive. Therapeutic vaccines are being considered as a possible way to combat this virus and more than 1,500 papers have been published to date aimed at creating these vaccines (https://pubmed.ncbi.nlm.nih.gov/?term=EBV+Vaccine&sort=date). The authors of the manuscript “A liposome-based nanoparticle vaccine induces effective immunity against EBV infection” set the task of creating a liposomal nanoform of a vaccine against EBV. The manuscript presents a new approach where viral antigens are covalently linked to the molecular components of the liposome membrane.
The Introduction contains "general" information about the virus and justifies the need to create a prophylactic vaccine (lines 57-58), emphasizing that there is none. The EBV glycoproteins are described in detail, some data on their use are given, and the formulation of the research problem is substantiated. The "age" of the articles used to describe the virus is puzzling - from 2006 to 2018. I think the reader will be interested in learning something new from the publications of the last 5 years. The authors "did not notice" or did not consider it necessary to note in the Introduction the development of nanovaccines against EBV, although 14 articles on this problem have been published.
There are questions regarding the Materials and Methods section. Thus, methods are described that prove the binding of purified recombinant protein gp350D123 to the surface of liposomes, however, methods for quantitative assessment of the degree of binding and methods for assessing the presence of unbound protein in the obtained Lipo-gp350D123 sample are not provided. Please clarify these points.
The section 2.6. Immunization assay does not contain information about immunization of mice with the Lipo-gp350D123 sample; data on "proteins" is provided (Line 154). This is unacceptable, the Materials and Methods should provide specific information about the methods used in all experiments. The missing data on the concentration of Lipo-gp350D123 for immunization is related to my previous question. Please make corrections.
The section 2.9. has a strange title: “Induction and purification of EBV-GFP reporter virus”. Please explain to me what "Induction" means in relation to virus? I recommend using the proper terms accepted in virology.
Please, clarify the origination and source of AKATA cells.
The Materials and Methods section needs careful editing: many sentences are constructed unusually and are difficult to understand.
The Results clearly and in detail present the production of liposomes and evidence of the localization of gp350D123 on the outer surface of the membrane. The illustrations provided are of high quality, consistent with the text, and help to understand the experiments and their results. There are no doubts about the reliability of the data presented. Please pay attention to the captions on the figures, there are errors (Fig. 3), perhaps more.
The work lacks a study of the protective properties of the new EBV vaccine variant on model animals. Such studies are presented in a number of published papers on EBV nanovaccines, positive results are reported. I think that the data on the effectiveness of the created liposomal vaccine variant in animals will serve as a weighty argument when comparing different nanovaccines against this harmful virus. I suggest that the authors conduct animal studies, judging by the presented work, they have such opportunities.
The authors have obtained an important result: immunization of mice with Lipo-gp350D123 ensures the production of neutralizing antibodies. However, the same results are reported in a dozen published articles on EBV nanovaccines. The creation of a new type of liposomal nanostructure is a very interesting result, but I recommend the authors to discuss the potential advantages of this type of possible vaccine compared to other EBV nanovaccines.
Comments on the Quality of English LanguageThe text, especially Materials and Methods, needs editing by a native English speaker.
Author Response
Reviewer 1:
The EBV virus, which inhabits the lymphocytes of more than 90% of the Earth's population, has been actively studied for more than 50 years, however, new features of its interaction with the human body and the pathogenesis of a wide range of diseases associated with this virus are constantly being identified. Lymphocytes are the primary target of the Epstein-Barr virus, so infection with it causes serious damage to the immune system and makes the creation of an effective preventive vaccine remains illusive. Therapeutic vaccines are being considered as a possible way to combat this virus and more than 1,500 papers have been published to date aimed at creating these vaccines (https://pubmed.ncbi.nlm.nih.gov/?term=EBV+Vaccine&sort=date). The authors of the manuscript “A liposome-based nanoparticle vaccine induces effective immunity against EBV infection” set the task of creating a liposomal nanoform of a vaccine against EBV. The manuscript presents a new approach where viral antigens are covalently linked to the molecular components of the liposome membrane.
Response: We sincerely appreciate the time and effort you have dedicated to reviewing our manuscript and providing constructive feedback. We have carefully addressed each of your comments and revised the manuscript accordingly. The reviewer’s comments are in italics, which are followed by our responses in blue.
- The Introduction contains "general" information about the virus and justifies the need to create a prophylactic vaccine (lines 57-58), emphasizing that there is none. The EBV glycoproteins are described in detail, some data on their use are given, and the formulation of the research problem is substantiated. The "age" of the articles used to describe the virus is puzzling - from 2006 to 2018. I think the reader will be interested in learning something new from the publications of the last 5 years. The authors "did not notice" or did not consider it necessary to note in the Introduction the development of nanovaccines against EBV, although 14 articles on this problem have been published.
Response: We thank the reviewer for highlighting this oversight. We have now updated the Introduction and added a paragraph to introduce the recent advancements of nanovaccines in EBV prophylactic vaccines, including EBV gp350-Ferritin vaccine, constructed by National Institute of Allergy and Infectious Diseases, the only EBV nanovaccine in clinical trials, and two mRNA vaccines of EBV constructed by Moderna in clinical trials, mRNA-1189 (gL, gH, gp42 and gp220) and mRNA-1195 (gLgH, gp42, gp220 and latent protein), according to the record in Clinical Trials.gov. These additions provide a more comprehensive background. Page 2, Lines 78-97.
- There are questions regarding the Materials and Methods section. Thus, methods are described that prove the binding of purified recombinant protein gp350D123 to the surface of liposomes, however, methods for quantitative assessment of the degree of binding and methods for assessing the presence of unbound protein in the obtained Lipo-gp350D123 sample are not provided. Please clarify these points.
Response: We apologize for the incomplete description. In the revised manuscript, we have added details in Section 2.4, as below:
Synthetic liposome was mixed with purified recombinant protein gp350D123 (1:10, molar ratio of gp350D123 and liposome), and then incubated overnight at 4℃ with continuous rotating. The mixture was further concentrated and purified by ultrafiltration and Superose 6 Increase 10/300 GL gel filtration column (GE healthcare). The concentration of the Liposome- gp350D123 conjugations was determined by BCA protein assay kit (Solarbio) and WB, and then they were stored at 4℃. The SEC elution profiles provided in Fig. 2A.
- The section 2.6. Immunization assay does not contain information about immunization of mice with the Lipo-gp350D123 sample; data on "proteins" is provided (Line 154). This is unacceptable, the Materials and Methods should provide specific information about the methods used in all experiments. The missing data on the concentration of Lipo-gp350D123 for immunization is related to my previous question. Please make corrections.
Response: We thank the reviewer for catching this. We have revised this Section as below:
Six-weeks-old special-pathogen-free (SPF) female Balb/c mice with eight per group were immunized by intramuscular injection (i.m.). Purified gp350D123 protein (about 15 μg) and Lipo-gp350D123 (protein equivalent, about 15 μg) were diluted in PBS and then injected at week 0 (prime), week 2 (boost 1) and week 4 (boost 2). The sera of all immunized mice were separately collected from week 0 to week 14 and stored at -80°C prior to use to detect the protein-specific IgG titer and neutralization antibody titer. Euthanasia was finally performed at week 14 to collect the major tissues of immunized mice, including lung, spleen, heart, liver, and kidney for histopathology analysis. The rest of spleen of each mouse was used for splenocytes isolation.
- The section 2.9. has a strange title: “Induction and purification of EBV-GFP reporter virus”. Please explain to me what "Induction" means in relation to virus? I recommend using the proper terms accepted in virology.
Response: We agree that the term was unclear. “Induction” referred to the reactivation of EBV from EBV-positive CNE2 cells. The section title has been revised to: “Production and purification of EBV-GFP reporter virus.”
- Please, clarify the origination and source of AKATA cells.
Response: We thank the reviewer for pointing this out, and we have added the Section 2.1 to introduce the cell lines used in this study, as below:
2.1. Cell culture
HEK293 cells were cultured in suspension medium (KOP293, KAIRUI BIOTECH) at 37℃ with 5% CO2 and shaking. CNE2 cells (EBV transformed human nasopharyngeal carcinoma cell line, CVCL_6889) and AKATA cells (EBV transformed Burkitt’s lymphoma cell line, CVCL_3080) were cultured in RPMI-1640 medium and supplemented with 10% fetal bovine serum (FBS) at 37 C with 5% CO2. Cell lines were tested for mycoplasma.
- The Materials and Methods section needs careful editing: many sentences are constructed unusually and are difficult to understand.
Response: We thank the reviewer for pointing this out. The entire section has been thoroughly edited for clarity and grammatical accuracy.
- The Results clearly and in detail present the production of liposomes and evidence of the localization of gp350D123 on the outer surface of the membrane. The illustrations provided are of high quality, consistent with the text, and help to understand the experiments and their results. There are no doubts about the reliability of the data presented. Please pay attention to the captions on the figures, there are errors (Fig. 3), perhaps more.
Response: We thank the reviewer for pointing this out. Fig. 3 has been corrected, and all figure captions were carefully checked.
- The work lacks a study of the protective properties of the new EBV vaccine variant on model animals. Such studies are presented in a number of published papers on EBV nanovaccines, positive results are reported. I think that the data on the effectiveness of the created liposomal vaccine variant in animals will serve as a weighty argument when comparing different nanovaccines against this harmful virus. I suggest that the authors conduct animal studies, judging by the presented work, they have such opportunities.
Response: We thank the reviewer for this suggestion to improve our study. While our current study focuses on vaccine design and proof-of-concept immunogenicity, we acknowledge the importance of in vivo protection assays and we have added a statement in the Discussion to emphasize this point (Page 12, Lines 459-463). Our future work will include challenge experiments in humanized mouse model and non-human primate model.
- The authors have obtained an important result: immunization of mice with Lipo-gp350D123 ensures the production of neutralizing antibodies. However, the same results are reported in a dozen published articles on EBV nanovaccines. The creation of a new type of liposomal nanostructure is a very interesting result, but I recommend the authors to discuss the potential advantages of this type of possible vaccine compared to other EBV nanovaccines.
Response: We thank the reviewer for this advice, and we have expanded the Discussion to highlight the key advantages of our liposome displaying strategy with other EBV nanovaccines, especially a similar study, in which the authors used liposome to package antigen proteins inside (PMID: 38906867). (Page 12, Lines 452-457)
Reviewer 2 Report
Comments and Suggestions for Authors
I have read the interesting manuscript by Ping Li et al. The manuscript is already well-written, highly detailed, and scientifically sound.
Below are my minor suggestions for improvement before publication:
- Introduction/discussion sections: Since various nanoparticle-based vaccines for EBV are already in clinical trials, please include more literature/references about them. Additionally, please shortly provide more details about their structural composition, advantages, and how your liposome-based vaccine could address their limitations.
- Material and methods/results sections: The TEM protocol (negative staining) and the results regarding the identification of the double-layer structure could be described in more detail.
- Results section: In Figure 1E, could you further elaborate on how you determined that Lipo5 provided the optimal binding? Is it possible to quantify the gp350D135 protein/lipo-protein complex ratio from the Native PAGE image?
Author Response
Reviewer 2:
I have read the interesting manuscript by Ping Li et al. The manuscript is already well-written, highly detailed, and scientifically sound.
Below are my minor suggestions for improvement before publication:
Response: We sincerely appreciate the time and effort you have dedicated to reviewing our manuscript and providing constructive feedback. We have carefully addressed each of your comments and revised the manuscript accordingly. The reviewer’s comments are in italics, which are followed by our responses in blue.
- Introduction/discussion sections: Since various nanoparticle-based vaccines for EBV are already in clinical trials, please include more literature/references about them. Additionally, please shortly provide more details about their structural composition, advantages, and how your liposome-based vaccine could address their limitations.
Response: We thank the reviewer for these suggestions, and we have expanded and updated more references about the development of nanovaccines and also clinical trials in Introduction/discussion sections (Page 2, Lines 78-96, and Page 12, Lines 439-444). And we also have expanded the Discussion to highlight the key advantages of our liposome displaying strategy with other EBV nanovaccines. Our displaying strategy could increase the molecular weight of antigens to enhance their stability, and the covalent antigen conjugation ensures precise spatial orientation for B-cell recognition. In addition, compared to other nanovaccines with protein derived fusion apparatus, such as ferritin and hepatitis B core antigen, liposomal formulations are cost-effective and easier to manufacture (Page 12, Line 454-458). These advantages showed the potential clinical translation of our strategy and vaccine.
- Material and methods/results sections: The TEM protocol (negative staining) and the results regarding the identification of the double-layer structure could be described in more detail.
Response: We thank the reviewer for pointing this out. The liposomes in our study showed a three-layer structure in TEM images. The Liposome nanoparticles had two thick bilayer structure, but a thin layer between them, thus the Liposomes showed the three layers structure. And we have revised our analysis and described them in details. (Page 7, Lines 297-304).
- Results section: In Figure 1E, could you further elaborate on how you determined that Lipo5 provided the optimal binding? Is it possible to quantify thegp350D135 protein/ lipo-protein complex ratio from the Native PAGE image?
Response: We thank the reviewer for this advice. We have described how we determined that Lipo5 provided the optimal binding in detail at Page 6 Lines 261-269. As shown in Fig. 1E, the mobility of gp350D123 protein in the gel was apparently retarded by the liposomes, indicating the formation of liposome-gp350D123 protein complex. And the Liposome 5 showed an optimum binding to gp350D123 protein with the less protein migrating into the gel under the condition of equal amount of protein, compared to other liposomes. Thus, we used Liposome 5 for the further vaccine construction. And for your advice to quantify the gp350D135 protein/ lipo-protein complex ratio from the Native PAGE image, it is difficult because the bands of the protein in the Native-PAGE gel are smearing, which may be caused by the glycosylation modification of gp350 protein or the insufficient combination of lipo-protein complex. Thus, we highlighted the unbound gp350D123 in Fig. 1E, to distinguish the binding effect of these lipo-protein complex. We hope this revision could help the reader to understand how we evaluate the binding effect.
Reviewer 3 Report
Comments and Suggestions for Authors
- Some awkward phrasing and minor grammatical errors should be corrected. Example:
Current: "EBV infected about 95% of global individuals..."
Suggested: "EBV infects approximately 95% of the global population..."
- The effectiveness of gp350-based vaccines should be contextualized with more recent clinical trial data.
- Provide additional explanation regarding the liposome conjugation process and its efficiency compared to other vaccine platforms.
- Expand on the potential clinical translation of this vaccine.
- Discuss future directions, such as multi-antigen nanoparticle formulations or human clinical trials.
- Some references need updating to reflect the latest research in EBV vaccine development.
Author Response
Reviewer 2:
Response: We sincerely appreciate the time and effort you have dedicated to reviewing our manuscript and providing constructive feedback. We have carefully addressed each of your comments and revised the manuscript accordingly. The reviewer’s comments are in italics, which are followed by our responses in blue.
- Some awkward phrasing and minor grammatical errors should be corrected. Example: Current: "EBV infected about 95% of global individuals..." Suggested: "EBV infects approximately 95% of the global population..."
Response: We thank the reviewer for pointing this out. All suggested grammatical revisions, including the example provided (e.g., changing “EBV infected” to “EBV infects approximately 95% of the global population”), have been incorporated throughout the manuscript.
- The effectiveness of gp350-based vaccines should be contextualized with more recent clinical trial data.
Response: We thank the reviewer for this suggestion. According to the record of EBV vaccine in Clinical Trials.gov, the EBV gp350-Ferritin vaccine constructed by National Institute of Allergy and Infectious Diseases is undergoing clinical phase 1, including NCT04645147 and NCT05683834. In addition, two mRNA vaccines of EBV constructed by Moderna, mRNA-1189 (gL, gH, gp42 and gp220) and mRNA-1195 (gLgH, gp42, gp220 and latent protein) are evaluated in clinical phase 1/2. And the mRNA-1189 was reported to complete the clinical trial (NCT05164094) recently, and the report showed a significant inhibit of viral load in saliva in mRNA-1195 immunized subjects, compared to Placebo-controlled. We have added this information in the Introduction section (Page 2, Lines 78–84) . These updates highlight the importance of gp350 and other glycoproteins in EBV vaccine design.
- Provide additional explanation regarding the liposome conjugation process and its efficiency compared to other vaccine platforms.
Response: We thank the reviewer for this suggestion. A detailed explanation of the liposome conjugation and purification process have been provided in the Materials and Methods section (Page 3, Lines 146-151). And as for the advice to provide conjugation efficiency compared to other vaccine platforms, we usually detected the size and structure of nanoparticle by dynamic light scattering (DLS) and Transmission electron microscopy (TEM), and the retention by size exclusion chromatography (SEC) to investigate the conjugation efficiency as other publications (PMID: 37848029, PMID: 35705092, PMID: 38906867). We have added the results of our nanovaccine in retention by size exclusion chromatography (SEC) in Fig. 2A and describe these results in detail.
- Expand on the potential clinical translation of this vaccine.
Response: We thank the reviewer for this advice, and we have expanded the Discussion to highlight the key advantages of our liposome displaying strategy with other EBV nanovaccines. Our displaying strategy could increase the molecular weight of antigens to enhance their stability, and the covalent antigen conjugation ensures precise spatial orientation for B-cell recognition. In addition, compared to other nanovaccines with protein derived fusion apparatus, such as ferritin and hepatitis B core antigen, liposomal formulations are cost-effective and easier to manufacture (Page 12, Lines 454-459). These advantages showed the potential clinical translation of our strategy and vaccine.
- Discuss future directions, such as multi-antigen nanoparticle formulations or human clinical trials.
Response: We thank the reviewer for this advice, and we have expanded the Discussion and Conclusion to discuss future work, including the development of multi-antigen nanoparticle formulations and the necessary of evaluation in EBV infected humanized mouse and non-human primate models. (Page 12, Lines 459-463 and Lines 466-471 )
- Some references need updating to reflect the latest research in EBV vaccine development.
Response: We thank the reviewer for pointing this out and we have expanded and updated more references about development of nanovaccines and also clinical trials.
Round 2
Reviewer 1 Report
Comments and Suggestions for Authors
The authors of the manuscript "A liposome-based nanoparticle vaccine induces effective immunity against EBV infection" conducted a thorough revision of the first version.
In my opinion, the new version presents the work consistently and in detail. The article has become more understandable and interesting.
I thank the authors for their comprehensive responses to my comments and for making the appropriate changes to the text.
I have no questions or comments about the article, and I recommend publishing it in the current version.
Author Response
Reviewer 1:
The authors of the manuscript "A liposome-based nanoparticle vaccine induces effective immunity against EBV infection" conducted a thorough revision of the first version.
In my opinion, the new version presents the work consistently and in detail. The article has become more understandable and interesting.
I thank the authors for their comprehensive responses to my comments and for making the appropriate changes to the text.
I have no questions or comments about the article, and I recommend publishing it in the current version.
Response: We sincerely appreciate the reviewer's thoughtful evaluation of our revised manuscript and their positive feedback regarding the improvements made. We are grateful for the time and effort dedicated to reviewing our work, as well as the constructive input that has strengthened the clarity and impact of the study.
We are pleased that the revised version addresses the concerns raised previously and effectively communicates the significance of our findings. We fully support the reviewer's recommendation for publication and confirm that the manuscript is ready for acceptance in its current form.
Thank you once again for your invaluable contributions to enhancing this work.